# Natural History of Hazelnut Allergy and Current Approach to Its Diagnosis and Treatment

**DOI:** 10.3390/children10030585

**Published:** 2023-03-18

**Authors:** Arianna Giannetti, Alessandro Ruggi, Giampaolo Ricci, Giuliana Giannì, Carlo Caffarelli

**Affiliations:** 1Paediatrics Unit, IRCCS Azienda Ospedaliero-Universitaria di Bologna, 40138 Bologna, Italy; 2Specialty School of Pediatrics, Alma Mater Studiorum, University of Bologna, 40138 Bologna, Italy; 3Department of Medical and Surgical Sciences (DIMEC), University of Bologna, 40138 Bologna, Italy; 4Clinica Pediatrica, Azienda Ospedaliero-Universitaria, Medicine and Surgery Department, Università di Parma, 43126 Parma, Italy

**Keywords:** hazelnut allergy, tree nut allergy, component resolved diagnosis, allergen immunotherapy, food allergy, children, Cor a 1, Cor a 9, Cor a 14

## Abstract

Hazelnut allergy is the most prevalent type of nut allergy in Europe, with symptoms that can range from mild, such as hives and itching, to severe, such as anaphylaxis, particularly in patients who are sensitized to highly stable allergens, such as storage proteins. Compared to other types of food allergies, allergies to tree nuts, including hazelnuts, tend to persist throughout life. Although symptoms can appear in early childhood, they often continue into adulthood, with a minority of cases improving during adolescence. Currently, there is no curative treatment available for hazelnut allergy, and patients must adhere to a restrictive diet and carry autoinjective epinephrine. However, oral allergen immunotherapy (AIT) is a promising treatment option. Patients can be categorized based on their risk for severe reactions using various clinical, in vivo, and in vitro tests, including component-resolved diagnosis and oral food challenge. This review aims to provide an overview of the current knowledge of the natural history of hazelnut allergy and new approaches for its diagnosis and management.

## 1. Introduction

Hazelnut is a common culprit of food-induced systemic allergic reactions and anaphylaxis in Europe, especially in young children [1,2]. It is part of the tree nut family, together with walnut, almond, cashew, pistachio, and pecan. In Southern Europe, nuts are the second most common trigger of food anaphylaxis after cow’s milk [3].

The tree that produces hazelnuts is the common hazel, *Corylus avellana*, which is a member of the birch family. Hazelnuts play an important role in the diet of many European countries, being often used as ingredients in complex preparations, as well as consumed independently, in toasted or raw form. They are an energy-dense food with high nutritional value. The regular addition of small quantities of nuts (including hazelnuts) to a balanced diet can lower cardiovascular risk, the risk of stroke, and diabetes mellitus type two. It can also reverse metabolic syndrome and has the potential to reduce cancer-related and all-cause mortality [4,5].

Hazelnut allergy can present in different ways depending on the individual’s sensitization. Individuals who are primarily sensitized to seasonal pollens may experience mild oropharyngeal symptoms when consuming hazelnuts, which is referred to as oral allergy syndrome (OAS). This is usually caused by IgE reactivity to panallergens such as pathogenesis related-10 (PR-10) found in hazelnuts and pollens, particularly in birch pollen [6,7]. In contrast, patients who are primarily sensitized to hazelnuts may experience systemic reactions or anaphylaxis upon ingestion. In these cases, the implicated hazelnut allergens are typically seed storage proteins (SSPs) and lipid transfer proteins (LTPs) [8]. However, there is a wide geographical variation in the relative frequency of cross-reactive and primary sensitization, with different molecular allergens playing different roles across Europe [9,10,11]. 

As with other tree nut allergies, hazelnut allergy usually first manifests during childhood and tends to persist throughout life [12,13]. Patients must follow a restrictive diet and carry autoinjective epinephrine. This, coupled with the risk of severe reactions after unintentional exposure, is a source of significant distress. Patients and their caregivers often develop food-related anxiety and fear of unfamiliar foods, with an impairment of health-related quality of life (HR-QoL) [14,15]. Flokstra-de Blok et al. [15] showed that HR-QoL in food-allergic adolescents and adults is more compromised than in diabetes mellitus. Thus, special care should be given to measuring HR-QoL with currently available validated scales in these patients. 

Overcoming food avoidance is difficult, as the rate of failure to reintroduce previously avoided foods, including hazelnuts and peanuts, can reach 25% after a negative food challenge, and both peanuts and hazelnuts are especially prone to failure of reintroduction [16]. 

The goal of this review is to present a narrative synthesis of the current insights on the natural history of hazelnut allergy and available strategies to modify it while informing the clinician on recent diagnostic considerations that play a role in clinical decision-making. A literature search was done using PubMed and the Cochrane Library. The search was conducted using as keywords “hazelnut” and “allergy”, filtering for articles published in the last eleven years (January 2012–December 2022). The search strings can be found in Appendix A. Initially, article titles and abstracts were screened, and only those deemed relevant to the topic were included, while duplicates were removed. Additional references were added during the full-text review of selected articles if they were deemed relevant and missed by the initial search.

## 2. Natural History of Hazelnut Allergy

### 2.1. Prevalence of Hazelnut Allergy and First Manifestations

Food allergy in school-aged children has an estimated prevalence of 1.4–3.8% [17]. Challenge-confirmed IgE-mediated tree nut allergy has a prevalence of less than 2%, while estimates for probable tree nut allergy are 0.05–4.9% [1]. Among nuts, hazelnut is the most frequent trigger of hypersensitivity-systemic reactions in Europe [1,3,17,18,19]. 

An age-based approach to hazelnut allergy is useful. Sensitization to hazelnuts is common in adults, especially in birch-endemic areas where cross-reactivity between PR-10 proteins of birch (Bet v 1) and PR-10 of hazelnut (Cor a 1) is the driving force [10]. In Central and Northern Europe, Cor a 1 IgE are detected in 60–90% of individuals with hazelnut sensitization [10]. Cross-sensitization is typically associated with mild OAS (itching and swelling of the tongue and lips). In school-age children living in the Mediterranean area, OAS is linked to sensitization to LTPs and PR-10 [20]. In younger children, the role of birch pollen cross-reactivity is of secondary importance, while primary sensitization to SSPs, i.e., Cor a 9 and Cor a 14, and LTPs, i.e., Cor a 8, is more common [10,21]; the latter is especially true in the Mediterranean area, with a prevalence ranging from 36 to 83% [10,22]. An American study [21] also showed that children under three years of age are predominantly sensitized to SSPs, while sensitization to the cross-reactive Cor a 1 is much more common in adults than in children. 

SSPs have a higher diagnostic value in predicting the risk of severe reactions [23]. Cosensitization to hazelnut and birch pollen makes the primary allergy less probable [24].

In primary hazelnut allergy, reactions can be severe and anaphylaxis is not uncommon [2,25,26] These reactions can also be fatal. Physical exertion after hazelnut intake can be necessary for developing anaphylaxis [27]. The first reaction can occur after the first known ingestion in early childhood, and in young children, severe reactions are more common [23,28]. Half of all children with nut allergies have an anaphylactic reaction as their first clinical manifestation, without a significant difference between types of nuts [18]. Characteristics of primary sensitization to hazelnuts and tree nuts are summarized in Table 1 [28,29,30,31,32,33]. 

### 2.2. Cross-Sensitization with Tree Nuts

As with other tree nut allergies, often patients show sensitization to more than one nut or peanuts, complicating their management. 

Co-sensitization to multiple nuts is described in several studies, with frequencies varying widely (12–96.7%) and increasing with the age of the studied population [34,35]. It should be noted that sensitization to multiple nuts does not mean that the patient develops clinical symptoms in each of them. It has been observed that, while most patients were sensitized to multiple tree nuts, more than half were allergic only to 1 or 2 nuts [36]. There is an amino acid sequence identity between the SSPs of some tree nuts and seeds which leads to IgE cross-reactivity in vitro and theoretically to clinical cross-reactivity. Special attention should be given to walnuts in hazelnut-allergic children. The sequence homology between the 2S albumins of hazelnuts (Cor a 14) and walnuts (Jug r 1) is 55% [37]. The walnut vicilin Jug r 6, belonging to the 7S globulin, showed a high level of cross-reactivity with nCor a 11 and hazelnut extract [38]. Accordingly, Villalta et al. [39] showed that patients who have a primary allergy either to walnuts or hazelnuts, with sensitization to their respective SSPs, are at risk of potentially severe reactions to both nuts, due to the cross-reactivity between 2S-albumins and legumins. Eizur et al. [40] found that walnut oral immunotherapy not only induces desensitization to walnuts but also cross-desensitization to pecan and hazelnuts in coallergic patients. This finding underscores the value of immunotherapy even in patients with multiple tree nut allergies. 

Peanut sensitization or peanut allergy is also frequent in hazelnut-allergic patients, even if peanuts and hazelnuts are not botanically similar. In 161 subjects with hazelnut allergy, Masthoff et al. [41] found that they were also sensitized to peanuts in 68% of cases with clinical hypersensitivity reactions in 45% of cases. This relation was not associated with IgE cross-reactivity to SSPs. So, its mechanism has yet to be fully elucidated.

### 2.3. Association with Atopic Dermatitis

The association of hazelnut allergy with allergic diseases including food allergies, asthma, atopic dermatitis, and allergic rhinitis is common [18]. Tree nut allergy is associated with early onset, persistent, and severe atopic disease [42]. 

Atopic dermatitis is a common finding in patients with symptoms related to hazel and birch pollen [43] and is also associated with primary hazelnut allergy. Atopic dermatitis is closely linked to food allergies. Positive skin tests for hazelnut can be found in patients who have never ingested the food [44]. Filaggrin mutations in patients with atopic dermatitis are a risk factor for severe food-related allergic reactions, and especially to peanuts and hazelnuts [45]. This class of mutations was also associated with many food allergies irrespective of atopic dermatitis by Kalb et al. [46], who also found that they are associated with the persistence of egg and milk allergies. 

### 2.4. Association with Asthma

In patients with food allergies, asthma is associated with a higher risk of severe reactions and anaphylaxis [47]. 

Rentzos et al. [48] found that asthmatic adults in Sweden most commonly reported hazelnut consumption as a cause of adverse reactions with a large proportion being also IgE sensitized either to hazelnut or to birch, or both. This underscores the previous statement that in adults, cross-reactivity between hazelnuts and birch is the main driving force of reported adverse reactions, with these sensitizations being more common in asthma. A recent study [49] showed that asthmatic individuals between 10 and 35 years of age with coexisting sensitization to peanut and hazelnut SSPs have higher levels of inflammatory markers with higher levels of a fraction of exhaled nitric oxide and blood eosinophil count.

### 2.5. Natural Course of Hazelnut Allergy

Primary hazelnut allergy manifests early in life and, as with other tree nut allergies, and tends to be persistent with a low chance of resolution [12,13,50]. This is in contrast with, for example, cow milk and hen’s egg allergy, in which many children become tolerant by school age. Fleischer [13] estimated that only approximately 10% of young patients may outgrow tree nut allergies. Hazelnut avoidance and the fear associated with it are not easy to overcome. After a negative food challenge, tolerated foods are often not reintroduced into the child’s diet [16]. Considering the persistence of primary hazelnut allergy, pharmacological attempts to modify its natural history (or at least the risk of severe reactions) are a priority, and the most promising results have been gained with oral and sublingual immunotherapy (see below).

## 3. Diagnostic Considerations

The diagnosis of hazelnut allergy [51] comprises the history of clinical hypersensitivity reactions following the consumption of hazelnuts, a positive hazelnut skin prick test (SPTs) response, and/or the detection of hazelnut-specific IgE antibodies (sIgE). A positive finding of SPTs and sIgE is not always necessary to establish a diagnosis. Even when SPTs or sIgE results are negative (although rarely), it is possible that the child has an allergy. The oral food challenge (OFC), especially the double-blind placebo-controlled food challenge (DBPCFC), is considered the gold standard to confirm the diagnosis [51]. However, it is a time and resource-consuming approach, especially in the case of patients with multiple tree nut allergies, Furthermore, OFC exposes the child to the risk of serious adverse reactions [52]. 

A positive SPT with extract or prick-by-prick with fresh food, as well as positive hazelnut sIgE, only show hazelnut sensitization. In the absence of a suggestive history of reactions upon hazelnut exposure or consumption, the clinician should not prescribe an elimination diet. A study by Erhard et al. [24] highlighted that in a population of children of mean age eight years, sensitization to hazelnut had a prevalence of 20%, with symptoms typical for primary hazelnut allergy occurring only in 1%. 

SPT is cheap and safe, however, the accuracy in diagnosing tree nut allergy is not widely studied. So far, there is no consensus on cut-offs for wheal size to predict the OFC outcome of hazelnut [51]. A 3-mm wheal diameter is considered positive for hazelnut sensitization, though this cut-off size has a low positive predicting value (PPV) [53]. Ho et al. [54] showed that the best results could be reached by using a diameter > 8 mm that was associated with a PPV > 95% for hazelnut, cashew, walnut, and sesame. Elizur et al. [36] also supported a cut-off of 8 mm, but it was associated with 64% sensitivity, 77% specificity, and only 38% PPV. Other studies [53,55] found that cutoffs between 7 and 7.5 mm had 58.8–62% sensitivity and 81–90% specificity. Masthoff et al. [56] found that the same cut-off was associated with a PPV of only 74%, while the PPV increased if a cut-off of >16 mm was considered. Overall, it appears that hazelnut SPT has low PPV [57]. Regarding negative predictive value (NPV), an SPT < 3 mm with commercial extracts was associated with an NPV of 100% in some studies [54,56]. At variance, other studies showed that NPV was 4% [58] or 53% [55] using commercial extracts and 15% using natural hazelnut [58]. SPT might be negative due to the low concentration of single allergens in the extract.

Food sIgE is an additional diagnostic test that can be helpful for identifying allergic patients. In children, the NPV of hazelnut sIgE < 0.35 kU/l was high, varying from 88% [55], 97% [59], and 100% [56]. No clear cutoff for identifying patients who develop a clinical reaction to hazelnut can be proposed [51,57,60,61]. The PPV of hazelnut sIgE ≥ 0.35 kU/l varied from 37% [59] to 57% [56]. The effect of combining sIgE levels with SPT size on predictive value is poor [56].

### 3.1. Component Resolved Diagnosis

Recent studies continue to widen our understanding of allergen components of hazelnut with a growing impact on clinical decision making. Component-resolved diagnosis (CRD) may distinguish different sensitization profiles and may help to recognize the primary allergy. It may also be useful for stratifying patients according to the risk of severe reactions. Ten hazelnut allergens have been biochemically identified and classified as proteins or glycoproteins and are named Cor a 1, Cor a 2, Cor a 8, Cor a 9, Cor a 10, Cor a 11, Cor a 12, Cor a 13, Cor a 14, and Cor a thaumatin-like protein (TLP). The commercially available hazelnut component-sIgE includes those to Cor a 1, a PR-10 that is sensitive to gastric digestion and heat labile [8], to Cor a 9 and to Cor a 14, that are both SSPs, and to Cor a 8, that is a LTP. Both SSPs and LTP are unaffected by high temperatures and digestive enzymes [62,63].

Cor a 1 sensitization is associated with clinical hypersensitivity reactions that are usually local and mild [8]. Toasted or cooked hazelnuts are usually tolerated. Cor a 1 sIgE has low PPV, NPV, sensitivity, and specificity, and is a poor discriminator for primary hazelnut allergy [64,65]. Children can develop Cor a1 sIgE as the result of cross-reactivity with PR-10 from birch or birch-related tree pollen. That is the primary sensitizer [10,66]. 

Cor 2 is also related to cross-reactivity with birch and many other plant pollens [10], however, further studies are necessary to clarify its clinical significance. 

Cor a 8 may be associated with systemic reactions and cross-reactions with the LTPs of other plants [67]. There is a correlation between IgE to Cor a 8 and IgE to LTPs in other foods, especially walnuts [10,68]. 

Cor a 9 is a 11S globulin and Cor a 14 is a 2S albumin. They have a very weak correlation to sIgE to pollen, in contrast with Cor a 1 and Cor a 2 [10]. Cor a 9 has significant homology to proteins found in peanuts and soya [69]. The sIgE to Cor a 9 and Cor a 14 are the most accurate components for a primary hazelnut allergy diagnosis and they are associated with a high risk of systemic reactions [10,59,65,70,71,72,73,74]. Cor a 14 seems to have even better diagnostic accuracy than Cor a 9 in predicting the risk for moderate-severe reactions to hazelnut [54,60,64,65]. Overall, NPVs of Cor a9 sIgE and Cor a 14 sIgE are high [64]. Combined IgE testing for Cor a 9 and Cor a 14 has shown a good (>90%) NPV for primary hazelnut allergy [65]. On the other hand, the PPVs of Cor a 9 sIgE and Cor a 14 sIgE are low [64]. 

Cor a 10 is a heat shock protein. Cor a 11 is a 7S globulin. 

Oleosins are a group of proteins that have been identified in different nuts. In hazelnuts, known oleosins are Cor a 12 and Cor a 13, with an unclear clinical significance [62]. A new hazelnut oleosin named Cor a 15 has recently been identified as a possible cause of reactions to hazelnuts in a subgroup of pediatric patients [75].

### 3.2. Basophil Activation Test

The basophil activation test (BAT) is an additional diagnostic tool showing promising results in hazelnut allergy diagnosis. It might be useful, especially in patients sensitized to multiple tree nuts, and to correctly discriminate between allergic and tolerant patients [36,76,77,78]. For example, Brandström et al. [78] found that the combination of BAT and CRD for primary hazelnut allergy had high sensitivity (100%) and specificity (>97% for Cor a 14, >94% for Cor a 9 sIgE) compared to a DBPCFC. This indicates that, even if they might not replace DBPCFCs as the gold standard, BAT may provide reliable diagnostic accuracy, without exposing children to the risks of DBPCFCs. However, BAT is not routinely used because availability is limited, and standardization is lacking.

## 4. Prevention and Treatment

The question of the possible reduction of food allergy development by introducing early allergenic food into an infant’s diet is still open [79,80]. There is currently no data on the prevention of hazelnut allergy by early introduction into weaning diets, though clinical trials are ongoing [81,82]. Hazelnut avoidance is the cornerstone of treatment. Being widely used in foods, unintended ingestion of hazelnuts is possible, although hazelnuts are included in the list of allergens that, according to EU Regulation No 1169/2011, must appear on food labels. This should be done with a clearly recognizable typeset, even if they are present in small quantities. Both parents and patients should be educated to correctly interpret labels and to treat severe reactions if they occur [83]. In patients with mild oropharyngeal symptoms associated with PR-10 sensitization, the patient can choose to ingest a small quantity of the culprit food. 

Oral second-generation nonsedating antihistamines and corticosteroids are used for treating allergic reactions, including skin manifestations, like urticaria or rash, rhinitis, and OAS. Patients at risk for anaphylactic reactions should be provided with a treatment plan and carry an emergency kit with an adrenaline autoinjector [84] (Figure 1).

### 4.1. Can We Alter the Natural History of Hazelnut Allergy?

#### 4.1.1. Allergen Immunotherapy

Allergen immunotherapy (AIT) involves the administration of increasing amounts of allergen extracts or products over long periods of time to reprogram the immune system response and achieve desensitization. AIT for inhalant allergens has proven to induce long-term tolerance [85]. AIT for food allergens has long been an experimental approach, with no standardized protocols or standardized extracts. Recently, following the results of the PALISADE study, an oral product for peanut allergy has been approved, called AR101 [86]. 

So far, only the Canadian Society of Allergy and Clinical Immunology guidelines [87] recommend AIT for any food while other scientific bodies recommend AIT only for peanut, egg, and cow’s milk [88]. The sublingual immunotherapy (SLIT) or the oral immunotherapy (OIT) is used due to safety concerns [85]. The dose should be gradually enhanced under strict medical surveillance with available personnel and equipment for treating a severe anaphylactic reaction. The tolerated dose should be maintained at home until the next increasing step. However, anaphylactic reactions are possible also during maintenance at home, sometimes requiring epinephrine administration. A recent study [89] showed that some factors, such as a previous reaction treated in an emergency department or a reaction treated with epinephrine during induction at the hospital can help predict the risk of severe home reactions to OIT. Most patients experiencing these reactions still achieved desensitization with immunotherapy. Studies focusing on the AIT of hazelnut allergies are reported in Table 2.

Enrique et al. [90] conducted a randomized, DBPC study to evaluate the efficacy and safety of SLIT with drops of a standardized hazelnut extract in forty-one adults (18-60 years of age), graded in five strengths (from a cumulative dose of 2 x 10^−11^ mg up to 119.51 mg). The buildup phase (four days) was performed in the hospital setting, while maintenance (five months) was performed at home. Food challenges were performed before and after 2–3 months of maintenance therapy, with an observed significant increase in the tolerance threshold (up to a level of protection against most unintended ingestions). The safety of SLIT was excellent. A followup study [91] was performed after a one-year treatment confirming the threshold increase that was previously observed. 

Moraly et al. [92] reported the results of a single-center retrospective study on 100 children (median age five years) with hazelnut allergy that received OIT. A DBPCFC was performed at the time of diagnosis and an OFC was performed after 6 months of OIT. The initial DBPCFC allowed to establish of the eliciting dose. For daily OIT, the authors used ground hazelnuts. Starting from 1/10th of the eliciting dose, they increased it monthly without exceeding half of the eliciting dose. After six months of treatment, 34% were successfully desensitized (i.e., they tolerated 1635 mg of hazelnut protein, corresponding to eight whole hazelnuts); the remaining patients still showed an increased threshold of tolerance, which granted protection from accidental exposures due to contamination. With longer therapy, the proportion of desensitized patients increased. Successful desensitization was associated with older age, smaller wheal diameter on hazelnut SPT, lower hazelnut sIgE level, and absence of cashew allergy. There was no observed association between comorbidities such as atopic dermatitis and asthma and the failure of desensitization. Interestingly, a higher level of Cor a 14 IgE was also not associated with desensitization failure. No severe reactions were associated with OIT. 

Sabouraud et al. [93] presented retrospective results on 70 patients (median age 10 years) treated with hazelnut OIT. Daily doses of cooked hazelnut were administered starting from 10% of their individualized target dose, and then progressively increased over the course of six months. They were evaluated after six months and after one year. Overall, after one year, 51.4% reached a tolerated dose of more than 120 mg of hazelnut proteins. While 51% of patients experienced mild side effects, 2.9% had severe systemic reactions during OIT, and 21.4% of children discontinued treatment, with approximately one in four developing an aversion to hazelnuts, which could limit long-term compliance. 

Scurlock et al. [94] conducted a prospective study investigating the efficacy of walnut OIT in patients allergic both to walnut and tree nuts, including hazelnuts. After 142 weeks of walnut OIT, an OFC was performed with walnuts as well as with tree nuts: 88% of patients reached desensitization for both. More studies are warranted, since these results are promising for patients with coallergies that might gain desensitization to multiple nuts even with single-nut OIT. 

Similar results were reached in the aforementioned study by Elizur et al. [40]. In this study, walnut OIT cross-desensitized patients to hazelnuts and pecan in the case of coallergy. 

#### 4.1.2. Other Approaches

AIT has been combined with different pharmacological agents, especially monoclonal antibodies targeting the TH2 pathways, to increase efficacy and safety and reduce the duration of treatment courses [85].

Omalizumab is an anti-IgE humanized monoclonal antibody used in the treatment of severe asthma and chronic idiopathic urticaria and has been used off label for the treatment of various allergic conditions [95]. It has shown some promise as an adjunct treatment to improve the efficacy and safety of allergen immunotherapy [96,97] (Table 2). 

A recent study [98] was performed on children treated with omalizumab for severe asthma and who had a concurrent food allergy, including hazelnuts. Food allergy was defined as a history of immediate reactions after food ingestion, with proven sensitization by skin prick (wheal at least 3 mm larger than positive control) and in vitro test (allergen-specific IgE level ≥ 0.35 kU/L). Patients with a history of OIT failure, but not currently receiving it, were included. This study showed that omalizumab was able not only to improve asthma symptoms’ control but also to increase the food allergen threshold, reducing dietary restrictions and with a positive impact on patient QoL. Omalizumab remains an expensive treatment, with restrictive criteria for its prescription; however, these studies show promising results for altering the natural course of hazelnut allergy.

Another option could be dupilumab, a monoclonal antibody that inhibits IL-4 and IL-13 signaling, but at the moment the evidence supporting its effect on food allergies is scarce [99,100,101]. In Italy, dupilumab is currently approved as an additional maintenance therapy in severe asthma starting from six years of age, for severe atopic dermatitis from six years of age, and for chronic rhinosinusitis with nasal polyps in adults. 

**Table 2 children-10-00585-t002:** Summary of studies for allergen immunotherapy and other approaches to hazelnut allergy.

Study, Year, Reference	Study Design	N° of Patients, Age	Intervention	Methods	Outcomes	Safety
Enrique, 2005 [90]	DBRCT	12 active vs. 11 placebo, 18–60 yrs	SLIT with standardized hazelnut extract (unit masses of major allergens Cor a 1 and Cor a 8) vs. placebo	Buildup over 4 days (hospital), maintenance 5 months (at home). DBPCFC at baseline and after 3 months of maintenance.	Mean ED increased from 2.29 g to 11.56 g	Local reactions 7.4% (itching), systemic reactions 0.2% (only during buildup, no epinephrine required)
Enrique, 2008 [91]	Followup study	Same participants	All participants continued SLIT with standardized hazelnut extract on maintenance dose	DBPCFC at baseline and after 1 year on maintenance	ED threshold increase confirmed (mean 14.57 g) and lower sIgE to hazelnut and lower IL-10	No systemic reactions.
Scurlock, 2017 [94]	Prospective cohort	8, median age 9 yrs	Walnut OIT in patients with allergy to walnut and to another TN (including hazelnut) **	OFC after 142 weeks of walnut OIT for both walnut and tree nuts.	88% of desensitization to both walnut and tree nuts	Not reported
Elizur, 2019 [40]	Prospective cohort	73 patients (55 active, 18 controls), median age 7.9 yrs	Walnut OIT in patients with or without co-allergy to pecan/hazelnut/cashew	Initial escalation of 4 days to establish the highest tolerated dose; buildup phase with monthly escalations in clinic, target 4000 mg walnut protein; after desensitization reached, maintenance with daily 1200 mg walnut protein.	Cross-desensitization to hazelnuts in 53% of patients co-allergic to hazelnuts.	85% mild reactions during buildup in the clinic, 73% at home. 20% required epinephrine use in the clinic. No adverse reactions to hazelnuts consumption in patients who were successfully cross desensitized.
Moraly, 2020 [92]	Retrospective	100, IQR 3–9 yrs	OIT with ground hazelnuts	Monthly buildup from 1/10th ED up to 50% ED. For those not desensitized at 6 mo, new dose buildup scheme. Maintenance with 416 mg hazelnut protein. OFC was performed after 6 mo on OIT and tolerance of 1635 mg of hazelnut protein defined desensitization.	6 mo: 34% desensitized	No severe reactions
Sabouraud, 2022 [93]	Retrospective	70, median age 10 years (IQR 6–13 yrs)	OIT with cooked hazelnuts	Buildup phase: individual dose definition based on OFC if possible. Daily dose of 10% of target, with progressive dose buildup over 6 mo. Maintenance dose was defined individually.	12 mo: 51% desensitized	51% mild, 2.9% severe reactions, 24% hazelnut aversion
Fiocchi, 2019 [98]	Real-life efficacy study (observational)	1, 9 years *	Omalizumab 0.016 mg/kg/IgE every 2 to 4 weeks for 4 months	OFC after 4 months of treatment	ED threshold increase from 13.8 mg to 35′328 mg, measurable (PedsQL questionnaire) improvement in HR-QoL	Well tolerated

DBRCT, Double-blind, randomized, placebo-controlled trial; IQR, Inter-Quartile Range; ED, eliciting dose; OIT, oral immunotherapy * the study included 15 patients with a median age of 12 years, but only one patient was allergic to hazelnut. ** it is not specified how many of the patients included in the study were specifically allergic to hazelnuts.

## 5. Conclusions

Hazelnut allergy is common, often persistent, and associated with the risk of severe systemic reactions and anaphylaxis. Distinguishing between primary hazelnut allergy and cross-reactions with pollens, especially birch, is crucial for accurate diagnosis and effective treatment. When combined with clinical history and skin tests, molecular diagnosis can assist in identifying patients at risk and reducing the number of unnecessary OFCs. Patients with sIgE to SSPs (Cor a 9 and Cor a 14) and LTP Cor a 8 are at higher risk of systemic reactions, while patients with sIgE to the cross-reactive Cor a 1 usually only show local reactions. 

A limitation of this approach is that the PPV and NPV of diagnostic tests vary according to the prevalence of the condition in the population. However, a clinical history not typical of pollen cross-reactivity and negative IgE tests (which have a high NPV) are helpful in diagnosing tolerance to hazelnut [68]. On the other hand, IgE test results may suggest performing the OFC in those patients with discrepancies between clinical history and allergy tests. 

The main goal of AIT is not to cure food allergy but to increase the tolerance of the patient to the culprit food, thus reducing the impact of a food allergy on quality of life, by reducing the risk of severe reactions upon accidental ingestion, as well as food-related anxiety. These are promising results in the field of hazelnut AIT alone or combined with monoclonal antibodies. However, evidence is still limited and there is no standard of care, with no data on long-term tolerance. There is a need for studies in this field that could provide good-quality evidence to establish a standardized protocol for hazelnut AIT.

## Figures and Tables

**Figure 1 children-10-00585-f001:**
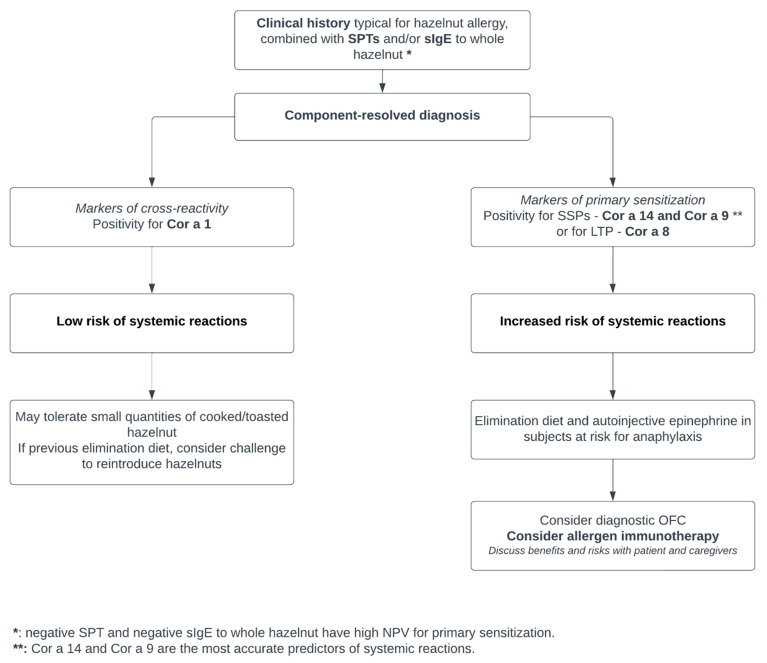
A possible algorithm for the diagnostic approach to hazelnut allergy and for identifying the potential candidates to allergen immunotherapy.

**Table 1 children-10-00585-t001:** Summary of studies reporting on the age of the first manifestation of hazelnut allergy.

Study, Year, Reference	Design	Countries	N° of Patients	Age of 1st Reaction	Severity
Tagliati, 2021 [18]	Retrospective	Italy	113 with confirmed nut allergy, 43 HA	3.7 ± 3.2 years	48% anaphylaxis *
Matias, 2020 [28]	Retrospective	Portugal	25 TN allergy (2 HA)	Mean age 3.1 years **	This study focused on TN anaphylaxis in preschoolers. There were no fatal events. 16% presented life-threatening glottis edema.
Cetinkaya, 2020 [30]	Prospective observational	Turkey	227 with TN and/or peanut allergy (63.9% HA)	Median age 9 mo. (range 6–12 mo.)	41.4% anaphylactic reactions after consumption of any TN or peanut
Clark, 2005 [31]	Prospective cross-sectional	United Kingdom	784 peanut or nut allergy, 319 HA	Median 2 years	28% moderately severe reaction (airway narrowing), 8% severe reactions
Sicherer, 2001 [33]	Voluntary registry	USA (5146), Canada (2), United Kingdom (1)	5149 registrants with peanut or TN allergy (<5% HA)	Median 36 mo., mean 77 mo.	1st reaction to TN was severe in ca. 30% of cases and a higher proportion of subsequent reactions was severe
Sicherer, 1998 [32]	Prospective observational	USA	122 (54 TN allergy, >5% HA)	Median age 62 mo. (range 10–264 mo.)	21.6% required epinephrine

HA = Hazelnut allergy. TN = Tree nut. * = This percentage refers to all patients included in the study, not only those with hazelnut allergy, but the occurrence and the severity of anaphylaxis at first reaction were not statistically different between the various nuts. ** = The study reports the mean age of the 1st anaphylactic reaction to TNs, that occurred after the 1st known ingestion of TN in 68% of cases.

## Data Availability

Not applicable.

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
