# Peer review of "Natural History of Hazelnut Allergy and Current Approach to Its Diagnosis and Treatment"

_children, 2023, doi:10.3390/children10030585_

Round 1

Reviewer 1 Report

The authors describe the current state of knowledge about tree nut allergy. Nut allergy is a particularly important clinical problem, not only because of its prevalence, but also because of the severity of its symptoms. Nut allergies affect both children and adults. Another contribution to the importance of the problem is the fact that many people already react to traces of peanuts with very severe symptoms, including anaphylactic shock.

The work is very well written. The authors reviewed the latest literature on nut allergy and presented the information obtained in a systematic, reliable and logical manner. It should be added that the literature on this subject is quite abundant, the authors managed to make a selection, present the most important facts in a very clear, concise and substantively impeccable way.

The authors discuss literature data on the prevalence of this type of allergy and pay attention to the diverse picture of allergies depending on the patient's age and whether it is primary sensitization or cross-reactivity. Separate chapters deal with the relationship between allergy to nuts and atopic dermatitis and asthma. After discussing the diagnostic problems described in the literature, the next chapter is devoted to molecular diagnostics, taking into account individual allergens. Such diagnosis determines the method of treatment and prognosis regarding the course of the disease. The authors present many papers devoted to the latest methods of treatment, including sublingual immunotherapy and immunotherapy with the use of monoclonal antibodies.

I believe that the work is very valuable and well written. The vast amount of information has been very well described, logically systematized, the article contains the most important, latest information that can be a valuable source for researchers.

Author Response

We sincerely thank the referee for the positive comment on our study.

Reviewer 2 Report

This is a well-written article that talks about a very important topic today, especially in pediatric medicine.

Only minor corrections to the English language and pronunciation are needed, along with a few additional corrections as listed below:

Line 25: In the abstract, an abbreviation is given without the full term at the first mention; OFC. The full term should be specified first

Line  74: It is stated that a period of ten years has been processed, and then the dates relating to a period of 11 years are given in parentheses. What is correct?

Line 115.-118. this paragraph would fit better in the chapter on prevention (3.3.) than in this part of the article where prevalence is discussed

Line 143:  expression at the end of the sentence: ‘..remains incompletely understood.’; expression is not common, find another one

Line 148 and 149: atopic dermatitis is mentioned first and then atopic eczema; to choose a name, to harmonize the expression

Line 167: the subsection of the article is marked incorrectly: after 3.1.4. it is stated in 3.1.6. ?

Line 179-181: this section does not clearly state the procedure for confirming the diagnosis of hazelnut allergy. Namely, it is necessary to clearly point out that even with negative findings of SPTs and sIgE (although rarely), it is possible that the child has an allergy. Therefore, a positive finding of SPTs and sIgE is not always necessary to establish a diagnosis. By definition, food allergy implies a reproducible clinical reaction even with negative in vivo and in vitro tests.

Line 208: typo;   should be corrected in ‘helpful’

Line 361: the title of the table must be understandable even outside the context of the article, the table must be independent. Therefore, the title cannot contain an abbreviation but the full name; Replace ‘AIT’ with the full name

Line 368: the title of the figure is unclear, unusual expressions are used, and grammatical correction is necessary

Line 370-372: as before, change the structure of the sentence, if necessary it can be divided into two sentences. A sentence structured in this way is difficult to understand, and difficult to follow to the end.

Author Response

This is a well-written article that talks about a very important topic today, especially in pediatric medicine.

Answer. We are grateful to the referee for the positive comment on our study.

Only minor corrections to the English language and pronunciation are needed, along with a few additional corrections as listed below:

Line 25: In the abstract, an abbreviation is given without the full term at the first mention; OFC. The full term should be specified first.

Answer. We thank the reviewer for the comment. The full term has been added.

Line  74: It is stated that a period of ten years has been processed, and then the dates relating to a period of 11 years are given in parentheses. What is correct?

Answer. We thank the reviewer that consent us to modify the length of the interval time.

Line 115.-118. this paragraph would fit better in the chapter on prevention (3.3.) than in this part of the article where prevalence is discussed.

Answer. As requested the paragraph has been moved to the prevention chapter. Accordingly, numbers of references have been revised.

Line 143:  expression at the end of the sentence: ‘..remains incompletely understood.’; expression is not common, find another one.

Answer. According to the reviewer’s remark, the expression has been changed.

Line 148 and 149: atopic dermatitis is mentioned first and then atopic eczema; to choose a name, to harmonize the expression.

Answer. As required, atopic eczema has been changed for atopic dermatitis.

Line 167: the subsection of the article is marked incorrectly: after 3.1.4. it is stated in 3.1.6. ?

Answer, 3.1.6 has been changed for 3.1.5

Line 179-181: this section does not clearly state the procedure for confirming the diagnosis of hazelnut allergy. Namely, it is necessary to clearly point out that even with negative findings of SPTs and sIgE (although rarely), it is possible that the child has an allergy. Therefore, a positive finding of SPTs and sIgE is not always necessary to establish a diagnosis. By definition, food allergy implies a reproducible clinical reaction even with negative in vivo and in vitro tests.

Answer. We understand the reviewer’s point of view. A sentence on this point has been added.

Line 208: typo;   should be corrected in ‘helpful’

Answer. Typos have been amended throughout the manuscript.

Line 361: the title of the table must be understandable even outside the context of the article, the table must be independent. Therefore, the title cannot contain an abbreviation but the full name; Replace ‘AIT’ with the full name.

Answer. Thank you for the criticism. The full name has been added.

Line 368: the title of the figure is unclear, unusual expressions are used, and grammatical correction is necessary.

Answer. Thank you for the criticism. The legenda has been changed.

Line 370-372: as before, change the structure of the sentence, if necessary it can be divided into two sentences. A sentence structured in this way is difficult to understand, and difficult to follow to the end.

Answer. We thank the referee for the comment. As requested, the sentence has been modified.

Reviewer 3 Report

This review aims to provide a valuable overview of the current knowledge of the natural history of hazelnut allergy and new approaches for its diagnosis and management. The manuscript is important as hazelnut allergy is a common and potentially severe allergic condition that may be associated with the risk of severe systemic reactions and anaphylaxis.

The Authors made an excellent effort to review current insights on the natural history of hazelnut allergy and available strategies to modify it. They also focused on  informing the clinicians about recent diagnostic considerations 

Methods a properly selected” The review presents a narrative synthesis of the current literature on hazelnut allergy. A literature search was conducted using PubMed and the Cochrane Library. 

The review documents a rather high prevalence of hazelnut allergy among school-aged children   (1.4-3.8%). The involvement of pan-allergens  (PR-10),  seed storage proteins (SSPs), and lipid transfer proteins (LTPs) is comprehensively discussed. What's important from the public health point of view is, special attention was given to walnuts in hazelnut-allergic children and peanuts.

The Authors made it very clear that although SPT is cheap and safe, its accuracy in diagnosing tree nut allergy is not confirmed. Possibilities for Allergen Immunotherapy (AIT) were well discussed.

The potential for therapy with AIT with different pharmacological agents, especially monoclonal antibodies has been reviewed  

Remarks:

Some typos” like e.g helpgul (line 208)

Author Response

Comment. Some typos” like e.g helpgul (line 208)
Answer. We sincerely thank the referee for the positive comment on our study. Typos have been amended throughout the manuscript.